# Reducing the Formation of Toxic Byproducts During the Photochemical Release of Epinephrine

**DOI:** 10.3390/jox15010008

**Published:** 2025-01-08

**Authors:** Mikhail A. Panfilov, Ezhena S. Starodubtseva, Tatyana Yu. Karogodina, Alexey Yu. Vorob’ev, Alexander E. Moskalensky

**Affiliations:** 1N.N. Vorozhtsov Novosibirsk Institute of Organic Chemistry SB RAS, 630090 Novosibirsk, Russia; 2Department of Physics, Novosibirsk State University, 630090 Novosibirsk, Russia

**Keywords:** epinephrine, adrenochrome, caged compounds, photopharmacology, platelet activation

## Abstract

Engineered light-sensitive molecules offer a sophisticated toolkit for the manipulation of biological systems with both spatial and temporal precision. Notably, artificial “caged” compounds can activate specific receptors solely in response to light exposure. However, the uncaging process can lead to the formation of potentially harmful byproducts. For example, the photochemical release of adrenaline (epinephrine) is accompanied by the formation of adrenochrome, which has neuro- and cardiotoxic effects. To investigate this effect in detail, we synthesized and compared two “caged” epinephrine analogs. The first was a classical compound featuring an *ortho*-nitrobenzyl protecting group attached to the amino group of epinephrine. The second analog retained the *ortho*-nitrobenzyl group but included an additional carbamate linker. The photolysis of both compounds was conducted under identical conditions, and the resulting products were analyzed using UV–Vis spectroscopy, chromatography, and NMR techniques. Surprisingly, while the classical compound led to the formation of adrenochrome, the carbamate-type caged epinephrine did not produce this byproduct, resulting in the clean release of the active substance. Subsequently, we assessed the novel compound in an in vitro platelet activation assay. The results demonstrated that the uncaging of epinephrine significantly enhances platelet activation, making it a valuable tool for advanced signaling studies.

## 1. Introduction

Engineering light-sensitive molecules allows researchers to precisely control biological activity using light. These molecules are not limited to genetically encoded photosensitive receptors but also include artificial compounds relying on photodissociation or photoisomerization [1]. These molecules can be designed to respond to specific wavelengths of light [2], triggering various cellular processes, such as receptor activation, gene expression [3], ion channel activity [4] or the inhibition of functional enzymes [5]. This approach not only enables complex signaling studies and enhances the understanding of biological systems [6] but also offers potential therapeutic applications constituting the emerging field of photopharmacology [7,8] and improvements in conventional photodynamic therapy [9,10].

In recent years, significant advancements have been achieved in developing drug delivery systems with enhanced abilities to selectively target anticancer agents to the tumor microenvironment, minimizing the damage to healthy tissue [11]. This goal is largely achieved through stimuli-responsive drug delivery systems, which can be classified into physical and chemical stimuli-responsive types [12]. Among the various stimuli used for controlled drug release, light stands out as a particularly compelling external trigger. Due to its on/off switching behavior, light allows for precise, remote-controlled drug release with exceptional spatial and temporal accuracy [13].

Adrenergic receptors are a class of G-protein-coupled receptors that mediate the physiological effects of catecholamines such as adrenaline (epinephrine) and noradrenaline (norepinephrine). These receptors are responsible for many physiological responses, including changes in heart rate, smooth muscle contraction, and metabolic processes. They are key targets for pharmacological agents: β-adrenergic agonists, for example, are used to treat asthma by activating β2-adrenergic receptors in airway smooth muscle to induce bronchodilation. Conversely, β-adrenergic antagonists, or beta-blockers, are employed to manage cardiovascular conditions like hypertension and arrhythmias. Agonists of α-adrenergic receptors, such as phenylephrine and oxymetazoline, are commonly used in nasal decongestants to constrict the blood vessels and reduce congestion. Additionally, epinephrine is critical in treating severe allergic reactions (anaphylaxis) and is also utilized in emergencies for cardiac arrest, croup, and asthma.

Given the diverse actions of adrenergic receptors, there is significant interest in developing light-sensitive compounds to manipulate these targets [14,15]. The initial work in this area was conducted by Muralidharan and Nerbonne [16], who introduced a family of photolabile “caged” adrenergic receptor agonists using the classical *ortho*-nitrobenzyl protecting group attached to the amino group of catecholamines. Recently, red-shifted photolabile protecting groups based on BODIPY were used for this purpose [17,18,19]. However, epinephrine is prone to photooxidation [20] and irreversibly forms adrenochrome, having neuro- and cardiotoxic effects [21,22], during uncaging [23]. Therefore, the above-mentioned studies were largely focused on phenylephrine or dopamine devoid of this drawback.

An additional carbamate linker has been introduced between the caging moiety and the cargo in the aforementioned BODIPY-based and other photocages [24]. This modification allows for the rapid cleavage of the carbamate bond connecting the catecholamine to the protecting group, leading to the release of the active compound. However, specific experiments with epinephrine have not been reported.

In this study, we aim at the design and synthesis of a photolabile “caged” analog of epinephrine with the diminished formation of adrenochrome during uncaging. We demonstrate that incorporating a carbamate linker between the *ortho*-nitrobenzyl protecting group and epinephrine allows us to achieve this goal, resulting in the clean release of the active compound. The described molecule is the first one that is capable of the uncaging of epinephrine that is not accompanied by unwanted byproducts. To demonstrate its biological application, we conducted experiments on platelet activation triggered by 365 nm UV light. The results indicate its strong potential to advance studies on platelet signaling.

## 2. Materials and Methods

### 2.1. Chemical Synthesis

Compound **1** (Figure 1) was synthesized as described in [16]. In brief, free epinephrine was treated with 2-nitrobenzyl bromide and potassium carbonate in dimethyl sulfoxide (DMSO). The NMR spectrum was consistent with the literature.

Synthesis of **2’ (2-nitrobenzyl (4-nitrophenyl) carbonate)**. First, 4-nitrophenyl chloroformate (0.514 g, 2.55 mmol, 1.5 eq) was dissolved in 10 mL of THF and cooled to 0 °C. Pyridine (0.190 mL, 2.55 mmol, 1.5 eq) was added to the reaction flask and stirred for 30 min under cooling conditions. Subsequently, a solution of 2-nitrobenzyl alcohol (0.260 g, 1.7 mmol, 1 eq) in 5 mL of THF was added dropwise over a 10 min period. The reaction was allowed to stir overnight at room temperature until completion, as indicated by TLC (DCM as the eluent). Then, 20 mL of DCM were subsequently added in order to dissolve the precipitate. The reaction mixture was transferred to a separation funnel and washed with 1M HCl (2 × 20 mL). The water phase was extracted with DCM, and the combined organic layers were dried over MgSO₄. The solvent was evaporated under reduced pressure, and the residue was recrystallized from 20 mL of petroleum ether–EtOAc (1:1). White solid, yield 0.429 g (79%).

The NMR spectrum was consistent with the literature.

1H NMR (300 MHz, Chloroform-d) δ 5.72 (s, 2H), 7.40 (d, J = 9.1 Hz, 2H), 7.52–7.60 (m, 1H), 7.68–7.74 (m, 2H), 8.18 (d, J = 8.2 Hz, 1H), 8.28 (d, J = 9.1 Hz, 2H).

Synthesis of **2 (2-nitrobenzyl (R)-(2-(3,4-dihydroxyphenyl)-2-hydroxyethyl)(methyl)carbamate)** (Figure 2). Experimental procedure was adopted from [25].

To a mixture of epinephrine (0.233 g, 1.275 mmol, 1.5 eq), HOBt (0.135 g, 0.85 mmol, 1 eq) and 4A molecular sieves (0.125 g) in DMF (5 mL) 1 (0.270 g, 0.85 mmol, 1 eq) were added. The reaction mixture was allowed to stir overnight at room temperature (16 h). After reaction completion, the molecular sieves were filtered off and washed with 5 mL of DMF. The filtrate was diluted with EtOAc (30 mL) and washed with brine a few times. The organic layer was dried over MgSO_4_ and the solvent was evaporated under reduced pressure. The obtained yellow oil was purified by column chromatography on silica gel (CHCl_3_:CH_3_CN). Yield 0.280 g (91%).

There were two sets of signals due to the presence of rotamers.

^1^H NMR (400 MHz, Chloroform-d; Appendix A) δ 2.85 (s, 3H), 3.29 (dt, J = 14.4, 8.1 Hz, 1H), 3.50 (dd, J = 14.1, 8.1 Hz, 1H), 4.08 (s, 1H), 4.68–4.75 (m, 1H), 5.41 (s, 2H), 6.53–6.68 (m, 2H), 6.77 (s, 1H), 7.27–7.44 (m, 2H), 7.53–7.61 (m, 1H), 7.96–8.01 (m, 1H).

^13^C NMR (101 MHz, Chloroform-d) δ 35.7, 56.7, 64.4, 72.3, 113.0, 115.1, 118.3, 124.8, 128.4, 128.5, 132.4, 133.4, 133.8, 143.7, 144.0, 147.0, 157.1.

HRMS: Due to the low intensity of the molecular ion of the corresponding caged epinephrine (**2’**), only the molecular ion of the dehydration product could be measured, whose mass was calculated (C_17_H_16_O_6_N_2_)^+^ to be 344.1003, found *m*/*z* = 344.1001.

### 2.2. Photolysis Experiments

The samples were irradiated by 350 nm LEDs (2 × 0.5 W; sample irradiance 1 W/cm^2^). The experiments were performed in phosphate-buffered saline (PBS) at pH 7.4, 1% DMSO, and the concentration of the compounds was 71 µM. UV–Vis absorption spectra were measured with a Shimadzu UV-1900 spectrophotometer between the irradiation sessions.

Additional experiments were performed with high-performance liquid chromatography (HPLC) using an Agilent 1260 Infinity II device equipped with a Poroshell column (120 EC-C18, 3 × 150 mm, 2.7 µm) and diode-array detector. In these experiments, the concentration of the compounds was 200 µM (H_2_O + 20% of acetonitrile). HPLC experiments were performed. The separation conditions were as follows: 98% water/2% ACN from 0 to 10 min; linear gradient to 100% ACN from 10 to 20 min; 100% ACN from 20 to 30 min. In these conditions, the retention time of the “caged” analogs of epinephrine was around 17 min, whereas the retention time of epinephrine was 1.5 min and the retention time of adrenochrome was nearly 2 min.

### 2.3. NMR Studies

First, 19.3 mg of 2 were dissolved in 4 mL of an ethanol + H_2_O mixture (9:1). The solution was flushed with argon in a quartz screw cap cuvette and placed under 365 nm light with air cooling. The mixture was irradiated for 1 h at room temperature, during which time there was no change in the reaction color. Then, the reaction mixture was transferred to a flask, and the solvent was evaporated under reduced pressure. CDCl_3_ was added to the residue, and the organic phase was transferred to an NMR tube.

Following the recording of the spectrum, it was observed that a solid had been separated, which was isolated by decantation and dissolved in DMSO-d6. The obtained NMR spectrum was found to correspond to that of epinephrine.

The same procedure was performed for compound **1**. In contrast to **2**, a dramatic change in color to cherry red was observed after irradiation. The obtained NMR spectrum of the precipitate resembled that of epinephrine but contained additional peaks.

### 2.4. Blood Platelet Activation Assay

The blood samples of healthy donors were obtained from the cubital vein. Written informed consent was obtained from the volunteers prior to the study. The study protocol was approved by the Ethics Committee of the Research Institute of Clinical and Experimental Lymphology—Branch of the Institute of Cytology and Genetics, Siberian Branch of the Russian Academy of Sciences.

The samples were collected in standard vacuum tubes containing sodium citrate as an anticoagulant (9:1). After this, the blood samples were settled at room temperature for about one hour until a layer of platelet-rich plasma was separated from the rest of the blood cells. Then, the fluorescent calcium probe Fluo-4-AM (Thermo Fisher Scientific, Waltham, MA, USA) was added to the samples for labeling. Next, 1 μL of Fluo-4-AM (1 mM stock solution in DMSO) was added to 61.5 μL of phosphate-buffered saline (PBS) and mixed with the blood plasma in 1 to 1 proportions. After 30 min incubation, 8 μL of the labeled cells were placed in the wells of an 8-well polystyrene strip plate with the addition of 3 μL of caged ADP (NPE-caged ADP, 10 mM stock solution) and/or 5 μL of caged epinephrine **2**; the samples were then supplemented with PBS to a final volume of 80 µL. After another 30 min, the samples were used in the experiment. An AxioVert.A1 inverted microscope with a 20× objective lens and an AxioCam 503 mono camera were used for fluorescence video recording. Photoactivation was performed with UV LEDs (340 nm).

### 2.5. Analysis of Calcium Signaling in Platelets

Single-cell fluorescence dynamics were obtained using the TrackMate software (v4.0.1) [26]. Then, we implemented the calcium spike detection algorithm using the Python environment. We used the LOWESS filter to smooth the data, as well as the find_peaks function from the Scipy library. The parameters of the function were set as follows: prominence = 1; minimum height = mean fluorescence intensity plus one standard deviation (to eliminate noise).

In essence, every detected platelet was assigned an array of fluorescence intensities and an array of found calcium spikes (locations and intensities). Then, we analyzed the platelets’ responses with the following metrics: the peak to peak distance in seconds; the difference between the maximum intensity and the mean value of the intensity; and the difference between the initial and final intensity values.

Furthermore, we separated the platelet populations in each experiment into 4 groups: non-activated (no spikes were detected); activated and deactivated before the flash (spikes detected only before the stimulus); activated after the flash (spikes detected only after the stimulus); always activated (spikes were detected throughout the whole experiment; this type of platelet response could involve spontaneous activation).

## 3. Results and Discussion

### 3.1. Photolysis of Caged Epinephrines

In this study, we synthesized and systematically compared two distinct “caged” epinephrine analogs designed for photochemical activation. The first analog was a classical caged compound in which an *ortho*-nitrobenzyl protecting group was attached to the amino group of epinephrine. This well-established strategy has been widely employed to control the release of bioactive molecules upon light exposure, providing a reliable mechanism for uncaging. The second analog retained the same *ortho*-nitrobenzyl group but introduced an additional carbamate linker between the protecting group and the epinephrine molecule. The inclusion of the carbamate group was intended to modify the photolysis process, potentially leading to the reduced photooxidation of the product.

Both compounds were subjected to photolysis under identical experimental conditions, allowing for a direct comparison of their photoreactivity and the efficiency of uncaging. The irradiation wavelength was 365 nm as it is relatively long and minimally absorbed by biological materials but still absorbed by target molecules. By using the same irradiation parameters and solvent conditions, we aimed to isolate the effects of the structural modifications on the photochemical behavior of the caged epinephrine. Additionally, a control study with free epinephrine was also performed. Epinephrine is also prone to photooxidation with the formation of adrenochrome [27].

Adrenochrome is a chemical compound that gained some attention in the middle of the XXth century as a potential cause of schizophrenia [28]. Some studies have reported the hallucinogenic and psychoactive properties of this compound. It has also been implicated in cardiotoxicity [22] and generally is a cytotoxic molecule [21]. Adrenochrome can be produced by the oxidation of epinephrine [29,30]. Recently, we showed that adrenochrome was produced during the photolysis of “caged” epinephrine analogs [23].

Figure 1A presents the absorption spectra of all compounds under study, each at the same concentration (71 µM) in PBS with 1% DMSO. While the compounds exhibit primary absorption maxima around 280 nm, the extinction coefficients of the “caged” analogs are significantly higher than that of free epinephrine. Upon photolysis, a new absorption band appears in the visible region of the spectrum, indicating the formation of oxidation products of epinephrine (Figure 1C–E). Notably, the absorbance at 485 nm—corresponding to the absorption maximum of adrenochrome—shows a marked increase (Figure 1B). This spectral feature is characteristic of adrenochrome formation, which is a key indicator of epinephrine oxidation.

Interestingly, while the classical ortho-nitrobenzyl “caged” epinephrine analog shows a substantial increase in absorbance at 485 nm, the carbamate-type analog exhibits a response similar to that of free epinephrine, suggesting that it does not undergo enhanced photooxidation during photolysis. This finding is significant as it implies that the carbamate modification may influence the rate or extent of oxidation, a hypothesis that warrants further investigation.

To validate this result and gain more insight into the photolysis products, we analyzed the photolysis mixture by high-performance liquid chromatography (HPLC). Figure 2 presents chromatograms of the photolysis products analyzed by HPLC, with the absorbance measured at 480 nm (left column) and 280 nm (right column). In the top row, the chromatogram of free epinephrine is shown (retention time ~1.5 min, marked by the blue band), while the bottom row shows the chromatogram of adrenochrome (retention time ~2 min, marked by the pink band).

The results indeed indicate the greater formation of epinephrine and reduced adrenochrome formation following photolysis in the carbamate-modified analog compared to the classical caged compound.

The lack of oxidation observed during the process can be attributed to the initial formation of carbamic acid as an intermediate during the photoremoval of the nitrobenzyl group (Figure 3). Carbamic acid is hypothesized to exhibit reduced reactivity toward oxidation by singlet oxygen, thereby preventing the formation of oxidation byproducts during this stage. Subsequently, the decarboxylation of carbamic acid occurs, resulting in the formation of epinephrine as the final product.

### 3.2. NMR Studies

To perform the NMR analysis, the concentrated mixtures of **1** and **2** in EtOH + H_2_O were irradiated for 1 h at room temperature. The color of the solution of **1** changed to cherry red, whereas there was no change in color for the solution of **2**.

The NMR analysis of a photolysis mixture is complicated because it contains residuals of the photoremovable protecting groups [31]. However, these residuals are soluble in chloroform, while epinephrine is not. Therefore, the mixture after photolysis was dried and dissolved in CDCl_3_, and the precipitate was dissolved in DMSO-d_6_.

The NMR analysis of the two isolated products revealed distinct differences in their compositions (Figure 3). The spectrum of the isolated product of **2** (middle row) shows clear and well-defined peaks corresponding to the characteristic proton signals of the epinephrine molecule (for which the spectrum is shown at the top). In contrast, the product of **1** displays only trace amounts of epinephrine, as evidenced by the significantly weaker and less resolved peaks in the same spectral regions. The spectrum also reveals additional peaks, corresponding to undesired byproducts of photolysis. These findings suggest that, while epinephrine is present in the second mixture, it is present in lower concentrations and with potentially toxic residuals. These results confirm the superiority of the novel carbamate-type “caged” epinephrine **2**.

### 3.3. Blood Platelet Activation

Blood platelets are essential for vascular integrity and wound healing. However, their dysfunction or abnormal levels can lead to health issues like bleeding disorders, thrombotic events, and complications in cardiovascular diseases and cancer. Studying platelet function in the lab can uncover disease mechanisms and improve diagnostics and the monitoring of patients’ platelet status. In laboratory testing, an activation stimulus is applied in vitro, and the response is recorded. Common stimuli include adenosine diphosphate (ADP) and epinephrine. We recently showed that using “caged” analogs of ADP enables light-induced activation and provides insights into platelet signaling [6,32].

Building on this concept, “caged” epinephrine analogs extend the potential for light-induced activation, allowing the further investigation of platelet responses. While epinephrine alone does not activate platelets [33], it lowers their activation threshold, making it a valuable tool in studying the synergistic effects of multiple agonists. In vivo, platelet activation is regulated by a complex balance of agonists and antagonists, with non-additive effects that cannot be easily predicted in single-agonist studies. Despite its significance, this interplay remains underexplored [34,35].

To explore these opportunities, we conducted the following experiments. Platelets were obtained from healthy volunteers and prepared for calcium imaging, as described in the Materials and Methods. Calcium signaling is essential for platelet activation, so it can be characterized by measuring the calcium responses. Four experimental conditions were investigated: (1) a control sample (no agonists), (2) a sample supplemented with “caged” ADP (375 µM), (3) a sample treated with carbamate-type “caged” epinephrine (4.4 µM), and (4) a combination of both agonists.

The samples were imaged under a microscope, and the fluorescence was recorded over time, saved as video files. Each sample was photoactivated with a short UV LED flash (10 ms) at 20 s after the start of the recording.

Figure 4A displays the changes in the mean fluorescence intensity (averaged over the entire field of view) over time, relative to photoactivation, for all samples. The control group exhibited a decline in fluorescence, while the samples treated with the agonists showed a marked increase in the fluorescence intensity. The most pronounced effect was observed in the combination of “caged” ADP and epinephrine.

Single-cell fluorescence dynamics were obtained using the TrackMate software [26]. TrackMate allows one to automatically detect cells and reconstruct the dependence of the fluorescence over time, even when cells are in motion, as is typical for platelets in these experiments. The tracking process is outlined in Figure 4B. Examples of single-cell calcium dynamics with different patterns are shown in Appendix A. We performed custom data analysis, as described in the Materials and Methods section, identifying each calcium spike. We provide some results quantifying the calcium signaling in Appendix A (Appendix A). Importantly, this procedure allows the classification of platelets into distinct groups based on their calcium dynamics: (1) non-activated cells with stable fluorescence; (2) spontaneously activated cells exhibiting calcium spikes before and after the UV flash; (3) cells that were activated before the stimulus but became non-activated after; and (4) cells that were non-activated but became activated after the flash. The latter subset of platelets can be classified as “flash-activated”. These cells displayed calcium spikes only following photoactivation, which indicates the applicability of this approach.

Interestingly, the majority of the platelets were spontaneously activated in the control sample, whereas the numbers of flash-activated and flash-deactivated platelets were approximately the same (Figure 4C). In contrast, samples treated with the “caged” agonists contained considerably more flash-activated cells—38% for caged epinephrine and 63% for caged ADP—compared to 19% in the control sample.

In the sample treated with both “caged” ADP and epinephrine, the fraction of flash-activated cells was 47%, surprisingly lower than that for caged ADP. However, a distinct population of platelets was observed (see Appendix A). Rather than exhibiting calcium spikes, these cells demonstrated a synchronous and irreversible increase in the fluorescence intensity. This behavior likely reflects more robust and sustained activation, consistent with dual-agonist stimulation [36]. In addition, we observed that some cells of this type were spread on the surface, did not have sharp boundaries and therefore were not detected by the tracking algorithm. This unique population likely contributed to the altered dynamics of the mean fluorescence intensity observed in the treatment with both “caged” agonists.

## 4. Conclusions

In this paper, we show that the introduction of a carbamate linker into a “caged” analog of epinephrine diminishes the product’s photooxidation during uncaging. It results in the clean production of epinephrine without the accompanying formation of adrenochrome and downstream products, which have neuro- and cardiotoxic effects. Subsequently, we demonstrate the applicability of the novel carbamate-type “caged” epinephrine for in vitro platelet activation. Our results demonstrate that the uncaging of epinephrine significantly enhances platelet activation, making it a valuable tool for advanced signaling studies.

## Data Availability

The original contributions presented in this study are included in the article/Appendix A. Further inquiries can be directed to the corresponding author.

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
