# Peer review of "Reducing the Formation of Toxic Byproducts During the Photochemical Release of Epinephrine"

_jox, 2025, doi:10.3390/jox15010008_

Round 1

Reviewer 1 Report

Comments and Suggestions for Authors

The manuscript "Eliminating the formation of toxic byproducts during photo- chemical release of Epinephrine" presents a new, original research study.

- Title can be more concrete. The cases tends an avoiding or prevention;

- Abstract: line 12, "For example, ..." the sentence is not finished.

- The mechanism of the observed action can be mention;

- Introduction: line 33: ... also the functional enzyme inhibition [Ref.]; improvement of the PDT efficiency [Refs.] (there are many studies);

- M&M: Instead of small letter "l" must be "L", 10 mL and etc.

- line 115: with fluence (power density) ... mW/cm2 and light dose of ... J/cm2;

- 2.4. Erythrocyte test is more popular;

- Results is better to be separated from Discussion;

- line 162: Synthesis is not new but is good to be presented with the citation(s); Scheme 1 is a Figure; 

- line 195: Fig. 1, B: If you performed parallel studies the SD must be added;

- The products mention with neuro and cardio toxicities with suitable Refs.

Author Response

The manuscript "Eliminating the formation of toxic byproducts during photo- chemical release of Epinephrine" presents a new, original research study.

Comment 1:

Title can be more concrete. The cases tends an avoiding or prevention.

Response 1:

We agree with the Reviewer that the word “Eliminating” is not the most suitable, and suggest changing it to “Reducing”.

Comment 2:

Abstract: line 12, "For example, ..." the sentence is not finished.

Response 2:

We have restructured the sentence as follows:

“For example, photochemical release of adrenaline (epinephrine) is accompanied by the formation of adrenochrome, which have neuro- and cardiotoxic effects.”

Comment 3:

The mechanism of the observed action can be mention.

Response 3:

We thank the reviewer for their suggestion to include a discussion of the mechanism of the observed action. To address this, we have incorporated a detailed explanation of the proposed mechanism in the revised manuscript.

Specifically, we hypothesize that the lack of oxidation observed during the process can be attributed to the initial formation of carbamic acid as an intermediate during the photoremoval of the nitrobenzyl group. Carbamic acid is hypothesized to exhibit reduced reactivity toward oxidation by singlet oxygen, thereby preventing the formation of oxidation byproducts during this stage. Subsequently, decarboxylation of carbamic acid occurs, resulting in the formation of epinephrine as the final product.

This proposed mechanism and the corresponding Scheme have been added to the end of Section 3.1 of the manuscript. We believe this addition enhances the clarity and depth of our findings.

Comment 4:

Introduction: line 33: ... also the functional enzyme inhibition [Ref.]; improvement of the PDT efficiency [Refs.] (there are many studies).

Response 4:

We have added these points to the 1st paragraph of the Introduction with the following references:

5. Negi, A.; Kesari, K.K.; Voisin-Chiret, A.S. Light-Activating PROTACs in Cancer: Chemical Design, Challenges, and Applications. Applied Sciences 2022, 12, 9674, doi:10.3390/app12199674.

9. Lazzarato, L.; Gazzano, E.; Blangetti, M.; Fraix, A.; Sodano, F.; Picone, G.M.; Fruttero, R.; Gasco, A.; Riganti, C.; Sortino, S. Combination of PDT and NOPDT with a Tailored BODIPY Derivative. Antioxidants 2019, 8, 531, doi:10.3390/antiox8110531.

10. Sarabando, S.N.; Palmeira, A.; Sousa, M.E.; Faustino, M.A.F.; Monteiro, C.J.P. Photomodulation Approaches to Overcome Antimicrobial Resistance. Pharmaceuticals 2023, 16, 682, doi:10.3390/ph16050682.

Comment 5:

M&M: Instead of small letter "l" must be "L", 10 mL and etc.

Response 5:

Fixed.

Comment 6:

line 115: with fluence (power density) ... mW/cm2 and light dose of ... J/cm2

Response 6:

We have added the irradiance: “sample irradiance 1 W/cm2

Comment 7:

2.4. Erythrocyte test is more popular

Response 7:

We thank the reviewer for pointing this out. Unfortunately, we do not have expertise in the study of red blood cells. Our laboratory is currently focused on research involving blood platelets, including a project on photoinduced platelet activation. As such, testing novel compounds on platelets aligns with our research interests. Platelet studies not only motivate the development of new chemical agents but also provide an ideal platform for their testing.

Comment 8:

Results is better to be separated from Discussion

Response 8:

We understand the suggestion to separate the Results from the Discussion section. However, we believe that presenting the results alongside the discussion allows for a more cohesive interpretation of the findings. This approach helps to provide immediate context and relevance to the data. We hope that this structure will be acceptable, but we are happy to consider any further suggestions or revisions that would improve the clarity of the paper.

Comment 9:

line 162: Synthesis is not new but is good to be presented with the citation(s); Scheme 1 is a Figure

Response 9:

We have added the following phrase describing the synthesis of 1: In brief, free epinephrine was treated with 2-nitrobenzyl bromide and potassium carbonate in dimethyl sulfoxide (DMSO).

Comment 10:

line 195: Fig. 1, B: If you performed parallel studies the SD must be added

Response 10:

Although experiments were done several times, and the effects observed were consistent across replicates, direct statistical comparison is not feasible because different time points were used. The experiment presented in the manuscript is representative of the overall findings and includes the largest number of time points.

Comment 11:

The products mention with neuro and cardio toxicities with suitable Refs.

Response 11:

We have added references at line 57:

21. Smythies, J.; Galzigna, L. The Oxidative Metabolism of Catecholamines in the Brain: A Review. Biochimica et Biophysica Acta (BBA) - General Subjects 1998, 1380, 159–162, doi:10.1016/S0304-4165(97)00131-1.

22. Behonick, G.S.; Novak, M.J.; Nealley, E.W.; Baskin, S.I. Toxicology Update: The Cardiotoxicity of the Oxidative Stress Metabolites of Catecholamines (Aminochromes). Journal of Applied Toxicology 2001, 21, S15–S22, doi:10.1002/jat.793.

Reviewer 2 Report

Comments and Suggestions for Authors

The paper, "Eliminating the formation of toxic byproducts during photochemical release of Epinephrine", provides a systematic study of novel carbamate-type caged epinephrine compounds and their application in controlled biological studies. The topic is relevant to the growing field of photopharmacology, but there are areas for improvement in clarity, experimental design, and broader applicability.

Comments

1.     The literature review portion emphasizes photolabile compounds but miss the broader comparisons with alternative non-photolabile methods. Include comparative methods to showcase the uniqueness of photolysis techniques.

2.     The objective of the study is scattered across the introduction. Clearly state the research goals in a concise section or paragraph.

3.     There is a lack of discussion on alternative wavelengths or intensities during photolysis experiments. Provide a rationale for choosing the 365 nm UV light and discuss if broader wavelength ranges were considered.

4.     The study has insufficient replicates and statistical analysis for the photolysis experiments. Incorporate additional replicates and statistical comparisons to strengthen the findings.

5.     Calcium signaling dynamics lack sufficient depth in their interpretation. Add statistical analysis to single-cell dynamics and explore the physiological implications in more detail.

6.     Improve the resolution of figures 3 and 4.

Author Response

The paper, "Eliminating the formation of toxic byproducts during photochemical release of Epinephrine", provides a systematic study of novel carbamate-type caged epinephrine compounds and their application in controlled biological studies. The topic is relevant to the growing field of photopharmacology, but there are areas for improvement in clarity, experimental design, and broader applicability.

Comment 1:

The literature review portion emphasizes photolabile compounds but miss the broader comparisons with alternative non-photolabile methods. Include comparative methods to showcase the uniqueness of photolysis techniques.

Response 1:

We thank the reviewer for pointing this out. We have added the following paragraph into the Introduction:

“In recent years, significant advancements have been made in developing drug delivery systems with enhanced ability to selectively target anticancer agents to the tumor microenvironment, minimizing damage to healthy tissues [11]. This goal is largely achieved through stimuli-responsive drug delivery systems, which can be classified into physical and chemical stimuli-responsive types [12]. Among the various stimuli used for con-trolled drug release, light stands out as a particularly compelling external trigger. Due to its on/off switching behavior, light allows for precise, remote-controlled drug release with exceptional spatial and temporal accuracy [13].”

11. Ezike, T.C.; Okpala, U.S.; Onoja, U.L.; Nwike, C.P.; Ezeako, E.C.; Okpara, O.J.; Okoroafor, C.C.; Eze, S.C.; Kalu, O.L.; Odoh, E.C.; et al. Advances in Drug Delivery Systems, Challenges and Future Directions. Heliyon 2023, 9, e17488, doi:10.1016/j.heliyon.2023.e17488.

12. Adepu, S.; Ramakrishna, S. Controlled Drug Delivery Systems: Current Status and Future Directions. Molecules 2021, 26, 5905, doi:10.3390/molecules26195905.

13. Rahim, M.A.; Jan, N.; Khan, S.; Shah, H.; Madni, A.; Khan, A.; Jabar, A.; Khan, S.; Elhissi, A.; Hussain, Z.; et al. Recent Advancements in Stimuli Responsive Drug Delivery Platforms for Active and Passive Cancer Targeting. Cancers (Basel) 2021, 13, 670, doi:10.3390/cancers13040670.

Comment 2:

The objective of the study is scattered across the introduction. Clearly state the research goals in a concise section or paragraph.

Response 2:

We thank the reviewer for pointing this out. We have modified the last paragraph in the Introduction and added the following sentence:

“In this study, we aim at the design and synthesis of the photolabile “caged” analogue of epinephrine with diminished formation of adrenochrome during uncaging.”

Comment 3:

There is a lack of discussion on alternative wavelengths or intensities during photolysis experiments. Provide a rationale for choosing the 365 nm UV light and discuss if broader wavelength ranges were considered.

Response 3:

In our study, we selected 365 nm wavelength by the following considerations:

  1. The corresponding UV light is absorbed by target molecules
  2. It is relatively long, which minimizes off-target effects such as absorption by cells and biomolecules.
  3. This wavelength is commonly used for photolysis studies, and there are convenient and cheap LED sources and lasers (3rd harmonic of the most popular Nd:YAG laser corresponds to 355 nm).

We also tested the photolysis at 275 nm, as this wavelength coinsides with the absorption mamumum of target compounds. However, this wavelength is absorbed also by free adrenaline, resulting in its accelerated photooxidation with and the formation of adrenochrome at a rate comparable to that for "caged epinephrine 1". The data can be found in the Supplementary materials of our recent paper: https://doi.org/10.1007/s43630-024-00665-9

We have now included a discussion on this rationale in the revised manuscript (line 183). We thank the reviewer for highlighting this important aspect.

Comment 4:

The study has insufficient replicates and statistical analysis for the photolysis experiments. Incorporate additional replicates and statistical comparisons to strengthen the findings.

Response 4:

We appreciate the Reviewer’s suggestion regarding the replicates and statistical analysis for the photolysis experiments. In fact, the experiments were done several times, and the effects observed were consistent across replicates. However, different time points were used in the experiments, and therefore direct statistical comparison across all conditions is not feasible. The experiment presented in the manuscript is representative of the overall findings and includes the largest number of time points.

Comment 5:

Calcium signaling dynamics lack sufficient depth in their interpretation. Add statistical analysis to single-cell dynamics and explore the physiological implications in more detail.

Response 5:

We thank the reviewer for their insightful comment regarding the interpretation of calcium signaling dynamics. To address this, we have conducted additional statistical analyses on single-cell calcium signaling dynamics. We constructed tailored algorithm for the detection of calcium spikes and analyzed key parameters such as fluorescence intensity changes, peak-to-peak intervals, and the percentage of activated cells following stimulation. These analyses have allowed us to quantify the variability and heterogeneity of calcium responses at the single-cell level.

Specifically, (1) we have added the details of the peak detection algorithm to the M&M section; (2) we have revised Figure 4 and section 3.2, added the quantitative information about the fraction of of flash-activated cells in each sample and the corresponding discussion; (3) we have incorporated additional results of statistical analysis, such as peak-to-peak intervals, into the Supplementary materials.

We believe these additions significantly enhance the depth and rigor of our study.

Comment 6:

Improve the resolution of figures 3 and 4.

Response 6:

We thank the reviewer for pointing this out. We have improved the resolution and made chemical structures in the Figure 3 better visible. Figure 4 has been revised significantly.